# Causal Inference with Noisy and Missing Covariates via Matrix Factorization

**Nathan Kallus**∗        **Xiaojie Mao**∗        **Madeleine Udell**∗

Cornell University
{kallus, xm77, udell}@cornell.edu

## Abstract

Valid causal inference in observational studies often requires controlling for confounders. However, in practice measurements of confounders may be noisy, and can lead to biased estimates of causal effects. We show that we can reduce bias induced by measurement noise using a large number of noisy measurements of the underlying confounders. We propose the use of matrix factorization to infer the confounders from noisy covariates. This flexible and principled framework adapts to missing values, accommodates a wide variety of data types, and can enhance a wide variety of causal inference methods. We bound the error for the induced average treatment effect estimator and show it is consistent in a linear regression setting, using Exponential Family Matrix Completion preprocessing. We demonstrate the effectiveness of the proposed procedure in numerical experiments with both synthetic data and real clinical data.

## 1 Introduction

Estimating the causal effect of an intervention is a fundamental goal across many domains. Examples include evaluating the effectiveness of recommender systems [1], identifying the effect of therapies on patients' health [2] and understanding the impact of compulsory schooling on earnings [3]. However, this task is notoriously difficult in observatonal studies due to the presence of confounders: variables that affect both the intervention and the outcomes. For example, intelligence level can influence both students' decisions regarding whether to go to college, and their earnings later on. Students who choose to go to college may have higher intelligence than those who do not. As a result, the observed increase in earnings associated with attending college is confounded with the effect of intelligence and thus cannot faithfully represent the causal effect of college education.

One standard way to avoid such confounding effect is to control for all confounders [4]. However, this solution poses practical difficulties. On the one hand, an exhaustive list of confounders is not known a priori, so investigators usually adjust for a large number of covariates for fear of missing important confounders. On the other hand, *measurement noise* may abound in the collected data: some confounder measurements may be contaminated with noise (e.g., data recording error), while other confounders may not be amenable to direct measurements and instead admit only proxy measurements. For example, we may use an IQ test score as a proxy for intelligence. It is well known that using proxies in place of the true confounders leads to biased causal effect estimates [5, 6, 7]. However, we show in a linear regression setting that the bias due to measurement noise can be effectively alleviated by using many proxies for the underlying confounders (Section 2.2). For example, in addition to IQ test score, we may also use coursework grades and other academic achievements to characterize the intelligence. Intuitively, using more proxies may allow for a more accurate reconstruction of the confounder and thus may facilitate more accurate causal inference.

---

∗Alphabetical order

Therefore, collecting a large number of covariates is beneficial for causal inference not only to avoid confounding effects but also to alleviate bias caused by measurement noise.

Although in the big-data era, collecting myriad covariates is easier than ever before, it is still challenging to use the collected noisy covariates in causal inference. On the one hand, data is inevitably contaminated with missing values, especially when we collect many covariates. Inaccurate imputation of these missing values may aggravate measurement noise. Moreover, missing value imputation can at most gauge the values of noisy covariates but inferring the latent confounders is the most critical for accurate causal inference. On the other hand, the large number of covariates may include heterogeneous data types (e.g., continuous, ordinal, categorical, etc.) that must be handled appropriately to exploit covariate information.

To address the aforementioned problems, we propose to use low rank matrix factorization as a principled approach to preprocess covariate matrices for causal inference. This preprocessing step infers the confounders for subsequent causal inference from partially observed noisy covariates. Investigators can thus collect more covariates to control for potential confounders and use more proxy variables to characterize the unmeasured traits of the subjects without being hindered by missing values. Moreover, matrix factorization preprocessing is a very general framework. It can adapt to a wide variety of data types and it can be seamlessly integrated with many causal inference techniques, e.g., regression adjustment, propensity score reweighting, matching [4]. Using matrix factorization as a preprocessing step makes the whole procedure modular and enables investigators to take advantage of existing packages for matrix factorization and causal inference.

We rigorously investigate the theoretical implication of the matrix factorization preprocessing with respect to causal effect estimation. We establish a convergence rate for the induced average treatment effect (ATE) estimator and show its consistency in a linear regression setting with Exponential Family Matrix Completion preprocessing [8]. In contrast to traditional applications of matrix factorization methods with matrix reconstruction as the end goal, our theoretical analysis validates matrix factorization as a preprocessing step for causal inference.

We evaluate the effectiveness of our proposed procedure on both synthetic datasets and a clinical dataset involving the mortality of twins born in the USA introduced by Louizos et al. [9]. We empirically demonstrate that matrix factorization can accurately estimate causal effects by inferring the latent confounders from a large number of noisy covariates. Moreover, matrix factorization preprocessing enhances the performance of many causal inference methods and is robust to the presence of missing values.

**Related work.** Our paper builds upon low rank matrix completion methods that have been successfully applied in many domains to recover data matrices from incomplete and noisy observations [10, 11, 12]. These methods are not only computationally efficient but also theoretically sound with provable guarantees [8, 13, 14, 15, 16, 17]. Moreover, matrix completion methods have been developed to accommodate heterogeneous data types prevalent in empirical studies by using a rich library of loss functions and penalties [18]. Recently, Athey et al. [19] use matrix completion methods to impute the unobservable counterfactual outcomes and estimate the ATE for panel data. In contrast, our paper focuses on measurement noise in the covariate matrix. Measurement noise has been considered in literature for a long time [5, 6, 20]. Kuroki and Pearl [21] and Miao et al. [22] show that causal effects are identifiable when the emission probabilities of proxies given confounders are known and satisfy an invertibility condition. In contrast, our method assumes a simpler proxy model (Figure 1) and provides a practical approach to carry out the estimation based on matrix factorization. Louizos et al. recently [9] propose to use Variational Autoencoder as a heuristic way to recover the latent confounders from multiple proxies. In contrast, matrix factorization methods, despite stronger parametric assumptions, address the problem of missing values simultaneously, require considerably less parameter tuning, and have theoretical justifications.

**Notation.** For two scalars $a, b \in \mathbb{R}$, denote $a \vee b = \max\{a, b\}$ and $a \wedge b = \min\{a, b\}$. For an positive integer $N$, we let $[N] = \{1, 2, \ldots, N\}$. For a set $\Omega$, $|\Omega|$ is the total number of elements in $\Omega$. For matrix $X \in \mathbb{R}^{N \times p}$, denote its singular values as $\sigma_1 \geq \sigma_2 \geq \cdots \geq \sigma_{N \wedge p} \geq 0$, and its smallest singular value as $\sigma_{\min}$. The spectral norm, nuclear norm, Frobenius norm and max norm of $X$ are defined as $\|X\| = \sigma_1$, $\|X\|_\star = \sum_{i=1}^{N \wedge p} \sigma_i$, $\|X\|_F = \sqrt{\sigma_1^2 + \cdots + \sigma_{N \wedge p}^2}$ and $\|X\|_{\max} = \max_{ij} |X_{ij}|$ respectively. The projection matrix for $X$ is defined as $P_X = X(X^\top X)^{-1} X^\top$. We use $\operatorname{col}(X)$ to denote the column space of $X$ and $\sigma(z)$ to denote the sigmoid function $1/(1 + \exp(-z))$.

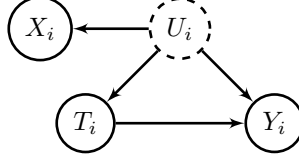

Figure 1: Causal graph for the $i$th individual, $i \in [N]$. The confounders $U_i$ are unobserved (dashed); the proxy variables $X_i$, treatment $T_i$, and outcome $Y_i$ are all observed (solid).

## 2   Causal inference with low rank matrix factorization

In this section, we first introduce the problem of causal inference under measurement noise and missing values formally and define notation. We then show that the bias caused by measurement noise in linear regression is alleviated when more covariates are used. Finally we review low rank matrix factorization methods and describe the proposed procedure for causal inference.

### 2.1   Problem formulation

We consider an observational study with $N$ subjects. For subject $i$, $T_i$ is the treatment variable and we assume $T_i \in \{0, 1\}$ for simplicity. We use $Y_i(0), Y_i(1)$ to denote the potential outcomes for subject $i$ under control and treatment respectively [4]. We can only observe the potential outcome corresponding to the received treatment level, i.e., $Y_i = Y_i(T_i)$. Assume that $\{Y_i(0), Y_i(1), T_i\}_{i=1}^{N}$ are independently and identically distributed (i.i.d). We denote $T = [T_1, ..., T_N]^{\top}$ and $Y = [Y_1, ..., Y_N]^{\top}$. For the ease of exposition, we focus on estimating the average treatment effect (ATE):

$$\tau = \mathbb{E}(Y_i(1) - Y_i(0)).$$

One standard way to estimate ATE is to adjust for the confounders. Suppose we have access to the confounders $U_i \in \mathbb{R}^r$ for subject $i$, $\forall i \in [N]$. Then we can employ many standard causal inference techniques (e.g., regression adjustment, propensity score reweighting, matching, etc.) to estimate ATE under the following unconfoundedness assumption:

**Assumption 1** (Unconfoundedness given unobservables). *For each $t = 0, 1$ and $i = 1, ..., N$, $Y_i(t)$ is independent of $T_i$ conditionally on $U_i$: $\mathbb{P}(T_i = 1 \mid Y_i(t), U_i) = \mathbb{P}(T_i = 1 \mid U_i)$.*

However, in practice we may not observe $\{U_i\}_{i=1}^{N}$ directly. Instead suppose we can only partially observe covariates $X_i \in \mathbb{R}^p$, which is a collection of noisy measurements for the confounders. The causal graph is given in Figure 1. The covariates $X_i$ can represent various data types by canonical encoding schemes. For example, Boolean data is encoded using 1 for true and $-1$ for false. Many other encoding examples, e.g., categorical data or ordinal data, can be found in Udell et al. [18]. We concatenate these covariates into $X \in \mathbb{R}^{N \times p}$. We assume that only entries of $X$ over a subset of indices $\Omega \subset [N] \times [p]$ are observed.

We further specify the generative model for individual entries $X_{ij}$, $(i, j) \in [N] \times [p]$. We assume that $X_{ij}$ are drawn indepedently from distributions $\mathbb{P}(X_{ij} \mid U_i^{\top} V_j)$, where $V_j \in \mathbb{R}^p$ represents loadings of the $j^{\text{th}}$ covariate on confounders. The distribution $\mathbb{P}(X_{ij} \mid U_i^{\top} V_j)$ models the *measurement noise* mechanism for $X_{ij}$. For example, if $X_{i1}$ is a measurement for $U_{i1}$ contaminated with standard Gaussian noise, then $\mathbb{P}(X_{i1} \mid U_i^{\top} V_1) \sim \mathcal{N}(U_i^{\top} V_1, 1)$ where $V_1 = [1, 0, ..., 0]^{\top}$. This generative model also accomodates *proxy variables*. Consider a simplified version of Spearman's measureable intelligence theory [23] where multiple kinds of test scores are used to characterize two kinds of (unobservable) intelligence: quantitative and verbal. Suppose that there are $p$ tests (e.g., Classics, Math, Music, etc.) which are recorded in $X_{i1}, ..., X_{ip}$ and the two intelligence are represented by $U_{i1}$ and $U_{i2}$. We assume that these proxy variables are noisy realizations of *linear combinations* of two intelligence. This can be modelled using the generative model $X_{ij} \sim \mathbb{P}(X_{ij} \mid U_i^{\top} V_j)$ with $V_j = [V_{i1}, V_{i2}, 0, ..., 0]^{\top}$ for $j \in [p]$. While this linear assumption seems restrictive, it's approximately true for a large class of nonlinear latent variable models when many proxies are used for a small number of latent variables [24].

We aim to estimate ATE based on $\mathcal{P}_{\Omega}(X)$, $Y$ and $T$. It is however very challenging for the presence of measurement noise and missing values. One the one hand, most causal inference techniques cannot

adapt to missing values directly and appropriate preprocessing is needed. On the other hand, it is well known that measurement noise can dramatically undermine the unconfoundedness assumption and lead to biased causal effect estimation [5, 6], i.e., $\mathbb{P}(Y_i(t)|T_i, X_i) \neq \mathbb{P}(Y_i(t)|X_i)$ for $t = 0, 1$.

## 2.2 Measurement noise and bias

In this subsection, we show that using a large number of noisy covariates can effectively alleviate the ATE estimation bias resulted from measurement noise in linear regression setting. Suppose there are no missing values. We consider the linear regression model: $\forall i \in [N]$, $Y_i = U_i^\top \alpha + \tau T_i + \epsilon_i$, where $\alpha \in \mathbb{R}^r$ is the coefficient for confounders $U_i$, $\tau$ is the ATE, and $\epsilon_i \overset{\text{i.i.d}}{\sim} \mathcal{N}(0, \sigma^2)$. For $\forall i \in [N]$, $T_i$ are independently and probabilistically assigned according to confounders $U_i$. Unconfoundedness (Assumption 1) implies that $T_i$ are independent with $\epsilon_i$ conditionally on $U_i$.

**Proposition 1.** *Consider the additive noise model: $X = UV^\top + W$ where $\{U_i\}_{i=1}^N$ are i.i.d samples, $W \in \mathbb{R}^{N \times p}$ contains independent noisy entries with mean 0 and variance $\sigma_w^2$, and entries in $W$ are independent with $\{U_i\}_{i=1}^N$. Suppose that $r$, $p$ are fixed and $p < N$. As $N \to \infty$, the asymptotic bias of least squares estimator in linear regression of $Y_i$ on $X_i$ and $T_i$ has the following form:*

$$\frac{\mathbb{E}(T_i U_i) \mathbb{E}(U_i^\top U_i)^{-1} [\frac{1}{\sigma_w^2} V^\top V + \mathbb{E}(U_i^\top U_i)^{-1}]^{-1} \alpha}{\mathbb{E}(T_i^2) - \mathbb{E}(T_i U_i)[(\frac{1}{\sigma_w^2} V^\top V)^{-1} + \mathbb{E}(U_i^\top U_i)]^{-1} \mathbb{E}(U_i^\top T_i)} \tag{1}$$

**Corollary 1.1.** *The asymptotic bias (1) diminishes to 0 when $\sigma_{\min}(V) \to \infty$.*

Corollary 1.1 suggests an important fact: collecting a large number of noisy covariates is an effective remedy for the bias induced by measurement noise, as long as the noisy covariates are sufficiently informative about the confounders. The condition $\sigma_{\min}(V) \to \infty$ requires that all confounders have asymptotically infinitely many proxies as $p \to \infty$.[2] Surprisingly, in this independent additive noise case, the asymptotic bias (1) is even nearly optimal: it is identical to the optimal asymptotic bias we would have if we knew the unobservable $V$ (Proposition 2, Appendix A). In the rest of the paper, we further exploit this fact by using matrix factorization preprocessing which adapts to missing values, heterogenenous data types and more general noise models.

## 2.3 Low rank matrix factorization preprocessing

In this paper, we propose to recover the latent confounders $\{U_i\}_{i=1}^N$ from noisy and incomplete observations of $X$ by using low rank matrix factorization methods, which rely on the assumption:

**Assumption 2** (Low Rank Matrix). *The full matrix $X$ is a noisy realization of a low rank matrix $\Phi \in \mathbb{R}^{N \times p}$ with rank $r \ll \min\{N, p\}$.*

In the context of causal inference, Assumption 2 corresponds to the surrogate-rich setting where many proxies are used for a small number of latent confounders. For example, we have access to IQ test scores, coursework grades, academic achievements and other proxies for the unobserved confounder intelligence. Under the generative model in section 2.1, Assumption 2 implies that $\Phi = UV^T$ where $U = [U_1, ..., U_N]^\top$ is the confounder matrix and $V = [V_1, ..., V_p]^T$ is the covariate loading matrix. Although this assumption is unverifiable, low rank structure is shown to pervade in many domains such as images [11], customer preferences [10], healthcare [12], etc. The recent work by Udell and Townsend [24] provides theoretical justifications that low rank structure arises naturally from a large class of latent variable models.

Moreover, low rank matrix factorization methods usually assume the *Missing Completely at Random* (MCAR) setting where the observed entries are sampled uniformly at random [8, 25].

**Assumption 3** (MCAR). *$\forall (i, j) \in \Omega$, $i \sim \text{uniform}([N])$ and $j \sim \text{uniform}([p])$ independently and the sampling is independent with the measurement noise.*

Our paper takes the Exponential Family Matrix Completion (EFMC) as a concrete example, which further assumes exponential family noise mechanism for the measurement noise [8].

**Assumption 4** (Natural Exponential Family). *Suppose that each entry $X_{ij}$ is drawn independently from the corresponding natural exponential family with $\Phi_{ij}$ as the natural parameter:*

$$\mathbb{P}(X_{ij}|\Phi_{ij}) = h(X_{ij}) \exp(X_{ij}\Phi_{ij} - G(\Phi_{ij}))$$

*where $h : \mathbb{R} \to \mathbb{R}$ is any function and $G : \mathbb{R} \to \mathbb{R}$ (called the log-partition function) is a strictly convex analytic function with $\nabla^2 G(u) \geq \mathrm{e}^{-\eta|u|}$ for some $\eta > 0$.*

Exponential family encompass a wide variety of distributions like Gaussian, Poisson, Bernoulli, which are extensively used for modelling different data types [26]. For example, if $X_{ij}$ takes binary values $\pm 1$, then we can model it using Bernoulli distribution: $\mathbb{P}(X_{ij} \mid \Phi_{ij}) = \sigma(X_{ij}\Phi_{ij})$.

EFMC estimates $\Phi$ by the following regularized *M*-estimator:

$$\hat{\Phi} = \min_{\|\Phi\|_{\max} \leq \frac{\alpha^*}{\sqrt{Np}}} \frac{Np}{|\Omega|} [\textstyle\sum_{(i,j)\in\Omega} - \log \mathbb{P}(X_{ij}|\Phi_{ij})] + \lambda \|\Phi\|_{\star} \tag{2}$$

The estimator in (2) involves solving a convex optimization problem, whose solution can be found efficiently by many off-the-shelf algorithms [27]. The nuclear norm regularization encourages a low-rank solution: the larger the tuning parameter $\lambda$, the smaller the rank of the solution $\hat{\Phi}$. In practice, $\lambda$ is usually selected by cross-validation. Moreover, the constraint $\|\Phi\|_{\max} \leq \frac{\alpha^*}{\sqrt{Np}}$ appears merely as an artifact of the proof and it is recommended to drop this constraint in practice [28]. It can be proved that under Assumptions $2 - 4$ and some regularity assumptions the relative reconstruction error of $\hat{\Phi}$ converges to 0 with high probability (Lemma 4, Appendix A). Furthermore, EFMC can be extended by using a rich library of loss functions and regularization functions [18, 29].

We use the left singular matrix of $\hat{\Phi}$ corresponding to nonzero singular values to estimate the *column space* of the confounder matrix $U$. Although $\hat{U}$ has orthonormal columns, the original confounders are allowed to be correlated (Assumption 5). The estimated confounder space matrix $\hat{U}$ is then used in place of the covariate matrix for subsequent causal inference methods (e.g., regression adjustment, propensity reweighting, matching, etc.). Admittedly, only the column space of the confounder matrix $U$ can be identified, and any nonsingular linear transformation of $\hat{U}$ is a valid estimator. However, this suffices for many causal inference techniques. For example, regression adjustment methods based on linear regression [7], polynomial regression, neural networks trained by backpropagation [30], propensity reweighting or propensity matching using propensity score estimated by logistic regressions, and Mahalanobis matching are invariant to nonsingular linear transformations. Moreover, the invariance to linear transformation and scale-free property is important since the latent confounders may be abstract without commonly acknowledged scale or units (e.g., intelligence).

## 3   Theoretical guarantee

In this section, we derive an error bound for the ATE estimator induced by EFMC preprocessing (2) in linear regression setting. Proofs are deferred to Appendix A.

Consider the linear regression model in Section 2.2. Suppose that we use EFMC preprocessing and linear regression for causal inference, which leads to the ATE estimator $\hat{\tau}$. It is well known that the accuracy of $\hat{\tau}$ relies on how well the estimated column space $\mathrm{col}(\hat{U})$ approximates the column space of true confounder matrix $\mathrm{col}(U)$. Ideally, if $\mathrm{col}(\hat{U})$ aligns with $\mathrm{col}(U)$ perfectly, then $\hat{\tau}$ is identical to the least squares estimator based on true confounders and is thus consistent. We introduce the following distance metric between two column spaces [31]:

**Definition 1.** *Consider two matrices $\hat{M} \in \mathbb{R}^{N \times k}$ and $M \in \mathbb{R}^{N \times r}$ with orthonormal columns, the principal angle between their column spaces is defined as*

$$\angle(M, \hat{M}) = \sqrt{1 - \sigma_{r \wedge k}^2(\hat{M}^\top M)}$$

This metric measures the magnitude of the "angle" between two column spaces. For example, $\angle(M, \hat{M}) = 0$ if $\mathrm{col}(M) = \mathrm{col}(\hat{M})$ while $\angle(M, \hat{M}) = 1$ if they are orthogonal.

**Theorem 1.** *Assume the following assumptions hold:*

(a) $\|\alpha\|_{\max} \leq A$ *for a positive constant $A$;*

(b) $\frac{1}{\sqrt{Nr}}\|U\|$ is bounded above for any $N$;

(c) $U_i$ is almost surely not linearly dependent with $T_i$;

(d) $r\angle(\hat{U}, U) \to 0$ as $N \to 0$;

(e) unconfoundedness (Assumption 1).

Then there exists a constant $c > 0$ such that with probability at least $1 - 2\exp(-cN^{1/2})$,

$$|\hat{\tau} - \tau^*| \leq \frac{(\frac{2A}{\sqrt{N}}\|T\|)(\frac{1}{\sqrt{Nr}}\|U\|)(r\angle(U, \hat{U}))}{\frac{1}{N}T^\top(I - P_U)T - \frac{2}{N}\|T\|^2\angle(U, \hat{U})} \xrightarrow{N \to \infty} 0 \tag{3}$$

In the above theorem, assumption (c) rules out multicollinearity between the treatment and the confounders, which is necessary for identifying ATE. This assumption guarantees that $\frac{1}{N}T^\top(I - P_U)T$ in (3) is bounded away from 0 for any $N$ (Lemma 5, Appendix A). Assumption (d) states that $\mathrm{col}(\hat{U})$ should converge to $\mathrm{col}(U)$ with rate faster than $1/r$ to guarantee consistency of the resulting ATE estimator. Theorem 1 shows that bounding the ATE estimation error requires bounding $\angle(U, \hat{U})$, i.e., the error of estimating $\mathrm{col}(U)$ in matrix factorization. In the following theorem, we derive an upper bound on the column space estimation error for EFMC (2).

**Theorem 2.** Assume that the following assumptions hold:

(a) Assumption 1 - 4 (Unconfoundedness, Low Rank Matrix, Missing Completely at Random, Natural Exponential Family);

(b) $X_{ij}$ is sub-Exponential conditionally on $U_i$ for any $(i, j)$;

(c) For $\Phi = UV^\top$, $\frac{\sigma_r(\Phi)}{\sigma_1(\Phi)}$ is bounded away from 0;

(d) $\hat{U}$ is estimated by EFMC (2) with $\lambda = 2c_0\sigma'\sqrt{Np}\sqrt{\frac{r\overline{N}\log\overline{N}}{|\Omega|}}$, where $\overline{N} = N \vee p$ and $|\Omega| > c_1 r\overline{N}\log\overline{N}$ for positive constants $c_0$ and $c_1$;

Then the following holds with probability at least $1 - 4e^{-2\log^2\overline{N}} - e^{-2\log\overline{N}}$:

$$\angle(\hat{U}, U) \leq \frac{c_2\alpha_{sp}(\Phi)\sqrt{\frac{r^3\overline{N}\log\overline{N}}{|\Omega|}}}{\frac{\sigma_r(\Phi)}{\sigma_1(\Phi)} - c_2\alpha_{sp}(\Phi)\sqrt{\frac{r^3\overline{N}\log\overline{N}}{|\Omega|}}} \wedge 1 \tag{4}$$

where $c_2 > 0$ is a constant and $\alpha_{sp}(\Phi) = \frac{\sqrt{Np}\|\Phi\|_{\max}}{\|\Phi\|_F}$ is the spikeness ratio of $\Phi = UV^\top$.

Theorem 2 shows that the column space estimation error of EFMC depends on two critical quantities: $\alpha_{sp}(\Phi)$ and $\frac{\sigma_r(\Phi)}{\sigma_1(\Phi)}$. The spikeness ratio $\alpha_{sp}(\Phi)$ is a standard measure quantifying the ill-posedness of matrix factorization problems [8, 32]. Small $\alpha_{sp}(\Phi)$ is necessary for accurate matrix estimation error for matrix factorization, i.e., small $\|\hat{\Phi} - \Phi\|$ (Lemma 6, Appendix A). Moreover, nonvanishing $\frac{\sigma_r(\Phi)}{\sigma_1(\Phi)}$ means that $\Phi$ does not lose information of any direction in $\mathrm{col}(U)$, and thus guarantees that small matrix estimation error $\|\hat{\Phi} - \Phi\|$ translates into small column space estimation error $\angle(\hat{U}, U)$ (Lemma 7, Appendix). Next we introduce some generative assumptions on confounder matrix $U$ and covariate loading matrix $V$. Under these assumptions, EFMC accurately estimates the column space of confounder matrix $U$ such that $r\angle(U, \hat{U}) \to 0$, and thus results in accurate ATE estimator.

**Assumption 5** (Latent Confounders and Covariate Loadings). $U$ and $V$ satisfy the following for some positive constants $\underline{v}$, $\overline{v}$, $c_V$ and $c_L$:

(a) For $i \in [N]$, $U_i$ are i.i.d Gaussian samples with covariance matrix $\Sigma_{r\times r} = LL^\top$ for some full rank matrix $L \in \mathbb{R}^{r\times r}$ such that $\frac{1}{\sqrt{r}}\|L\| < c_L$;

(b) $\underline{v}p \leq \sigma_r^2(VL^\top) \leq \sigma_1^2(VL^\top) \leq \overline{v}p$ and $\frac{\max_j\|V_j\|}{\|V\|_F} \leq \frac{c_V}{\sqrt{p}}$, $j = 1, ..., p$.

Assumption 5(a) specifies a Gaussian distribution for latent confounders, which implies assumption (b) in Theorem 1 with high probability (Lemma 10, Appendix A). It also assumes without loss of generality that the latent confounders are not perfectly linearly correlated. Moreover, Assumption 5(b) exludes the case where almost all covariates have vanishing loadings on the latent confounders. [3] In this case, the collected covariates are not informative enough for recovering the latent confounders.

**Theorem 3.** *Suppose that $r/N \to 0$ and $\exists \delta > 0$ such that $p^{1+\delta}/N \to 0$. Under the Assumption 5, there exist positive constants $c_3 - c_5$ such that*

- $\alpha_{sp}(\Phi) \leq c_3 c_V \sqrt{r \vee \log N}$ *with probability at least* $1 - N^{-1/2} - 2\exp(-c_4 N^{1/2})$;

- $\frac{\sigma_r(\Phi)}{\sigma_1(\Phi)} \geq \sqrt{\frac{v}{\underline{v} + 2\overline{v}}}$ *with high probability* $1 - 2\exp(-c_5 p^\delta)$;

*If we further assume the assumptions in Theorem 2 and that $\|\alpha\|_{\max} \leq A$, then for a constant $C > 0$,*

$$|\hat{\tau} - \tau^*| \leq \frac{2AC\sqrt{\frac{r^5 \overline{r} \overline{N} \log \overline{N}}{|\Omega|}}}{\frac{1}{N} T^\top (I - P_U) T \left[\sqrt{\frac{v}{\underline{v} + 2\overline{v}}} - \Lambda(r, \overline{N}, |\Omega|)\right] - 2\Lambda(r, \overline{N}, |\Omega|)},$$

*where $\Lambda(r, \overline{N}, |\Omega|) = C\sqrt{\frac{\overline{r} r^3 \overline{N} \log \overline{N}}{|\Omega|}}$ and $\overline{r} = r \vee \log N$.*

The assumption that $p^{1+\delta}/N \to 0$ appears as an artifact of proof and our simulation shows that the consistency also holds when $N < p$ (Figure 3, Appendix B). Theorem 3 guarantees that the ATE estimator induced by EFMC is consistent as long as $r^5 \overline{r} \overline{N} \log \overline{N}/|\Omega| \to 0$ when $N, p \to \infty$. This seems much more restrictive than consistent matrix reconstruction that merely requires $r\overline{N} \log \overline{N}/|\Omega| \to 0$ (Lemma 6, Appendix A). However, this is due to the pessimistic nature of the error bound. Our simulations in Section 4.1 show that matrix factorization works very well for $r = 5$, $N = 1500$ and $p = 1450$ such that $r^6 \gg N$.

## 4 Numerical results

In this section, we show that low rank matrix factorization effectively reduces the ATE estimation error caused by measurement noise using two experimental settings: 1) synthetic datasets with both continuous and binary covariates and 2) the twins dataset introduced by Louizos et al. [9]. To implement matrix factorization, we use the following nonconvex formulation:

$$\hat{U}, \hat{V} = \underset{U \in \mathbb{R}^{N \times k}, V \in \mathbb{R}^{p \times k}}{\operatorname{argmin}} \sum_{(i,j) \in \Omega} L_{i,j}(X_{ij}, U_i^\top V_j) + \frac{\lambda}{2}(\|U\|_F + \|V\|_F) \tag{5}$$

where $L_{ij}$ is a loss function assessing how well $U_i^\top V_j$ fits the observation $X_{ij}$ for $(i, j) \in \Omega$. The solution $\hat{U}$ is an estimator for the confounder space. This nonconvex formulation (5) is proved to equivalently recover the solution of the convex formulation (2) when log-likelihood loss functions and sufficient large $k$ are used [18, 28]. Solving the nonconvex formulation (5) approximately is usually much faster than solving the convex counterpart. In our experiments, we use the the R package softImpute [33] for continuous covariates and quadratic loss, the R package logisticPCA [34] for binary covariates and logistic loss, and the Julia package LowRankModels [18] for categorical variables and multinomial loss. All tuning parameters are chosen via 5-fold cross-validation.

### 4.1 Synthetic experiment

We generate synthetic samples according to the following linear regression process: $Y_i \mid U_i, T_i \sim \mathcal{N}(\alpha^\top U_i + \tau T_i, 1)$ where confounder $U_{ij} \sim \mathcal{N}(0, 1)$ and treatment variable $T_i \mid U_i \sim \text{Bernoulli}(\sigma(\beta^\top U_i))$ for $i \in [N]$, $j \in [r]$. We consider covariates generated from both indepedent Gaussian noise and independent Bernoulli noise: $X_{ij} \sim \mathcal{N}(U_i^\top V_j, 5)$ and $X_{ij} \sim$

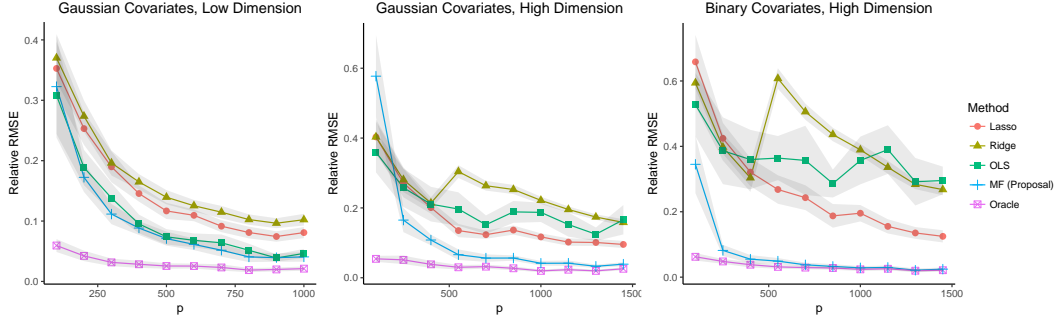

Figure 2: Estimation error for different ATE estimators with Gaussian and Binary proxy variables .

Bernoulli$(\sigma(U_i^\top V_j))$ for $V_j \in \mathbb{R}^r$. We set the dimension of the latent confounders $r = 5$, use $\alpha = [-2, 3, -2, -3, -2]$ and $\beta = [1, 2, 2, 2, 2]$, and choose $\tau = 2$ in our example. (But our conclusion is robust to different values of these parameter.) We consider low dimensional case where the number of covariates $p$ varies from 100 to 1000 and the sample size $N = 2p$ and high dimensional case where $p$ varies from 150 to 1500 and $N = p + 50$. For each dimensional setting, we compute the error metrics based on 50 replications of the experiments and we generate entries of $V$ independently from standard normal distribution with $V$ fixed across the replications.

We compare the root mean squared error (RMSE) scaled by the true ATE in Figure 2 for the following five ATE estimators in linear regression: the Lasso, Ridge and OLS estimators from regressing $Y_i$ on $T_i$ and noisy covariates $X_i$, the OLS estimator from regressing $Y_i$ on $T_i$ and the estimated confounders $\hat{U}_i$ from matrix factorization (MF), and the OLS estimator from regressing $Y_i$ on $T_i$ and the true confounders $U_i$ (Oracle). The shaded area corresponds to the 2-standard-deviation error band for the estimated relative RMSE across 50 replications.

Figure 2 shows that OLS leads to accurate ATE estimation for Gaussian additive noise when the number of covariates is sufficiently large, which is consistent with Corollary 1.1. However, for high dimensional data, matrix factorization preprocessing dominates all other feasible methods and its RMSE is very close to the oracle regression for sufficiently large number of covariates. While all feasible methods tend to have better performance when more covariates are available, matrix factorization preprocessing is the most effective in exploiting the noisy covariates for accurate causal inference. Sufficiently many noisy covariates are very important for accurate ATE estimation in the presence of measurement noise. We can show that the error does not converge when only $N$ grows but $p$ is fixed (Figure 6, Appendix B). With only a few covariates, matrix factorization preprocessing may have high error because the cross-validation chooses rank smaller than the ground truth. Furthermore, the gain from using matrix factorization is more dramatic for binary covariates, which demonstrates the advantage of matrix factorization preprocessing with loss functions adapting to the data types. More numerical results on different dimensional settings and missing data can be found in Appendix.

## 4.2 Twin mortality

We further examine the effectiveness of matrix factorization preprocessing using the twins dataset introduced by Louizos et al. [9]. This dataset includes information for $N = 11984$ pairs of twins of same sex who were born in the USA between 1998-1991 and weighted less than 2kg. For the $i^{\text{th}}$ twin-pair, the treatment variable $T_i$ corresponds to being the heavier twin and the outcomes $Y_i(0), Y_i(1)$ are the mortality in the first year after they were born. We have outcome records for both twins and view them as two potential outcomes for the treatment variable. Therefore, the $-2.5\%$ difference between the average mortality rate of heavier twins and that of ligher twins can be viewed as the "true" ATE. This dataset also includes other 46 covariates relating to the parents, the pregnancy and birth for each pair of twins. More details about the dataset can be found in Louizos et al. [9].

To simulate confounders in observational studies, we follow the practice in Louizos et al. [9] and selectively hide one of the two twins based on one variable highly correlated with the outcome: GESTAT10, the number of gestation weeks prior to the birth. This is an ordinal variable with values from 0 to 9 indicating less than 20 gestation weeks, $20 - 27$ gestation weeks and so on. We simulate $T_i \mid U_i \sim \text{Bernoulli}(\sigma(5(U_i/10 - 0.1)))$, where $U_i$ is the confounder GESTAT10. Then for each

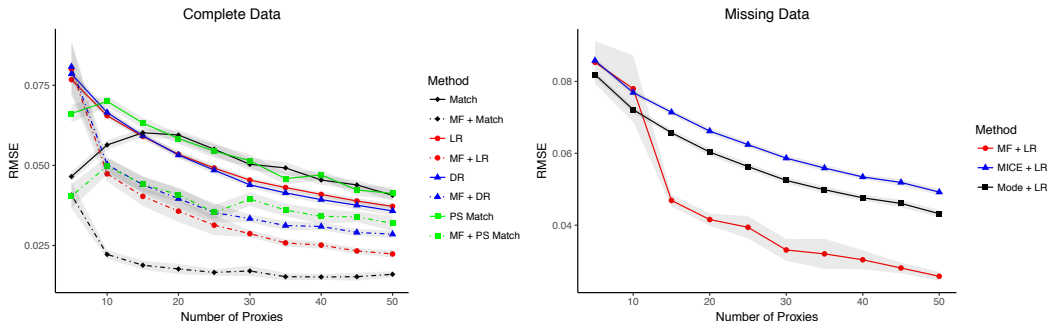

Figure 3: Left: estimation errors of different ATE estimators with no missing values. Right: estimation error of ATE estimators based on matrix factorization and imputation methods when each entry of the proxy variable matrix is missing with probability 30%.

twin-pair, we only observe the lighter twin if $T_i = 0$ and the heavier twin otherwise. We create noisy proxies for the confounder as follows: we replicate the GESTAT10 $p$ times and independently perturb the entries of these $p$ copies with probability $0.5$. Each perturbed entry is assigned with a new value sampled from 0 to 9 uniformly at random. We also consider the presence of missing values: we set each entry as missing value independently with probability $0.3$. We vary $p$ from 5 to 50 and for each $p$ we repeat the experiments 20 times for computing error metrics.

We compare the performance of different methods for both complete data and missing data in Figure 3. For complete data, we consider logistic regression (LR), doubly robust estimator (DR), Mahalanobis matching (Match) and propensity score matching (PS Match) using noisy covariates, and their counterparts using the estimated confounders from matrix factorization. All propensity scores are estimated by logistic regression using noisy covariates or estimated confounders accordingly. The matching methods are implemented via the full match algorithm in the R package optmatch [35]. For missing data, we consider logistic regression using data output from different preprocessing method: imputing missing values by column-wise mode, multiple imputation using the R package MICE with 5 repeated imputations [36], and the estimated confounders $\{\hat{U}_i\}_{i=1}^N$ from matrix factorization. We also discuss comparisons to [9] in Appendix C.

We can observe that all methods that use matrix factorization clearly outperform their counterparts that do not, even though the noise mechanism does not obey common noise assumptions in matrix factorization literature. In particular, the Mahalanobis matching (Match) benefits the most from matrix factorization that simultaneously alleviates the measurement noise and reduces the dimension. The effect of solely reducing measurement noise is shown in the result of the propensity score matching where matching is based on the univariate propensity score and thus dimensionality is not the primary issue. Our results also demonstrate that matrix factorization preprocessing can augment popular causal inference methods beyond linear regression. Furthermore, matrix factorization preprocessing is robust to a considerable amount of missing values and it dominates both the ad-hoc mode imputation method and the state-of-art multiple imputation method. This suggests that inferring the latent confounders is more important for causal inference than imputing the noisy covariates.

## 5    Conclusion

In this paper, we address the problem of measurement noise prevalent in causal inference. We show that with a large number of noisy proxies, we can reduce the bias resulting from measurement noise by using matrix factorization preprocessing to infer latent confounders. We guarantee the effectiveness of this approach in a linear regression setting, and show its effectiveness numerically on both synthetic and real clinical datasets. These results demonstrate that preprocessing by matrix factorization to infer latent confounders has a number of advantages: it can accommodate a wide variety of data types, ensures robustness to missing values, and can improve causal effect estimation when used in conjunction with a wide variety of causal inference methods. As such, matrix factorization allows more principled and accurate estimation of causal effects from observational data.

**Acknowledgments**

This work was supported by the National Science Foundation under Grant No. 1656996. This work was supported by the DARPA Award FA8750-17-2-0101.

## Footnotes

[2]For example, suppose the number of confounders is $r = 2$ and $V = \begin{bmatrix} 1 & 0 & 0 & \dots & 0 \\ 0 & 1 & 1 & \dots & 1 \end{bmatrix}^\top$. Then only the first covariate is a noisy proxy for the first confounder and $\sigma_{\min}(V) = 1 < \infty$ for any $p$.

[3] For example, if only $n_V$ noisy covariates have nonvanishing loadings on the confounders, and their loading vectors have norms of similar order, then $\|V\|_F = \sqrt{\sum_{j=1}^p \|V_j\|^2} \approx \sqrt{n_V} \max_j \|V_j\|$, so $\frac{\max_j \|V_j\|}{\|V\|_F} \approx \frac{1}{\sqrt{n_V}}$. When $\lim_{p \to \infty} n_V/p \to 0$, i.e., almost all covariates have vanishing loading, Assumption 5(b) is violated.

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
