[Supplementary Material]

# A  Proofs

## A.1  Measurement Noise and Bias

*Proof of Proposition 1.*

$$\hat{\tau} = [\frac{1}{N}T^\top(I - P_X)T]^{-1}[\frac{1}{N}T^\top(I - P_X)Y] \tag{6}$$

By Law of Large Number,

$$\frac{1}{n}T^\top X \to \mathbb{E}[T_i(U_iV^\top + W_i)] = \mathbb{E}(T_iU_i)V^\top$$

$$\frac{1}{N}X^\top Y \to \mathbb{E}[(U_iV^\top + W_i)^\top(U_i\alpha + \tau T_i + \epsilon_i)] = V\mathbb{E}[U_i^\top U_i]\alpha + \tau V\mathbb{E}(U_i^\top T_i)$$

$$\frac{1}{N}T^\top Y \to \mathbb{E}[T_i(U_i\alpha + \tau T_i + \epsilon_i)] = \mathbb{E}(T_iU_i)\alpha + \tau\mathbb{E}(T_i^2)$$

$$(\frac{1}{N}X^\top X)^{-1} \to [\mathbb{E}(VU_i^\top U_iV^\top + W_i^\top U_iV^\top + VU_i^\top W_i + W_i^\top W_i)]^{-1} = [V\mathbb{E}(U_i^\top U_i)V^\top + \sigma_w^2 I_{r\times r}]^{-1}$$

By Sherman–Morrison–Woodbury formula,

$$[V\mathbb{E}(U_i^\top U_i)V^\top + \sigma_w^2 I_{r\times r}]^{-1} = \frac{1}{\sigma_w^2}I - \frac{1}{\sigma_w^2}V[(\frac{1}{\sigma_w^2}\mathbb{E}U_i^\top U_i)^{-1} + V^\top V]^{-1}V^\top$$

and

$$[(\frac{1}{\sigma_w^2}\mathbb{E}U_i^\top U_i)^{-1} + V^\top V]^{-1} = (V^\top V)^{-1} - (V^\top V)^{-1}[(V^\top V)^{-1} + \frac{1}{\sigma_w^2}\mathbb{E}U_i^\top U_i]^{-1}(V^\top V)^{-1}$$

Plug these terms back in 6,

$$\begin{aligned}
\frac{1}{N}T^\top(I - P_X)Y &= \tau\mathbb{E}(T_i^2) + \mathbb{E}(T_iU_i)\alpha \\
&\quad + \mathbb{E}(T_iU_i)V^\top[V(\mathbb{E}U_i^\top U_i)^{-1}V^\top + \sigma_w^2 I]^{-1}V\mathbb{E}U_i^\top U_i\alpha \\
&\quad + \tau\mathbb{E}(T_iU_i)V^\top[V(\mathbb{E}U_i^\top U_i)^{-1}V^\top + \sigma_w^2 I]^{-1}V^t op\mathbb{E}U_i^\top T_i \\
&= \tau\mathbb{E}(T_i^2) + \mathbb{E}(T_iU_i)\alpha \\
&\quad + \frac{1}{\sigma_w^2}\mathbb{E}(T_iU_i)\{V^\top V[(\frac{1}{\sigma_w^2}\mathbb{E}U_i^\top U_i)^{-1} + V^\top V]^{-1}V^\top V - V^\top V\}\mathbb{E}(U_i^\top U_i)\alpha \\
&\quad + \frac{\tau}{\sigma_w^2}\mathbb{E}(T_iU_i)\{V^\top V[(\frac{1}{\sigma_w^2}\mathbb{E}U_i^\top U_i)^{-1} + V^\top V]^{-1}V^\top V - V^\top V\}\mathbb{E}(U_i^\top T_i) \\
&= \tau\mathbb{E}(T_i^2) + \mathbb{E}(T_iU_i)\alpha - \frac{1}{\sigma_w^2}\mathbb{E}(T_iU_i)\{\frac{1}{\sigma_w^2}\mathbb{E}(U_i^\top U_i) + (V^\top V)^{-1}\}^{-1}E(U_i^\top U_i)\alpha \\
&\quad - \frac{\tau}{\sigma_w^2}E(T_iU_i)\{\frac{1}{\sigma_w^2}E(U_i^\top U_i) + (V^\top V)^{-1}\}^{-1}E(U_i^\top T_i)
\end{aligned}$$

Similarly,

$$\begin{aligned}
\frac{1}{N}T^\top(I - P_X)T &\to \mathbb{E}(T_i^2) - \mathbb{E}(T_iX_i)(\mathbb{E}X_i^\top X_i)^{-1}\mathbb{E}(X_i^\top T_i) \\
&= \mathbb{E}(T_i^2) - \frac{1}{\sigma_w^2}\mathbb{E}(T_iU_i)\{\frac{1}{\sigma_w^2}\mathbb{E}(U_i^\top U_i) + (V^\top V)^{-1}\}^{-1}\mathbb{E}(U_i^\top T_i)
\end{aligned}$$

Therefore,

$$\begin{aligned}
\hat{\tau} - \tau &\to \frac{\mathbb{E}(T_iU_i)\alpha - \frac{1}{\sigma_w^2}\mathbb{E}(T_iU_i)\{\frac{1}{\sigma_w^2}\mathbb{E}(U_i^\top U_i) + (V^\top V)^{-1}\}^{-1}E(U_i^\top U_i)\alpha}{\mathbb{E}(T_i^2) - \frac{1}{\sigma_w^2}\mathbb{E}(T_iU_i)\{\frac{1}{\sigma_w^2}\mathbb{E}(U_i^\top U_i) + (V^\top V)^{-1}\}^{-1}\mathbb{E}(U_i^\top T_i)} \\
&= \frac{\mathbb{E}(T_iU_i)\mathbb{E}(U_i^\top U_i)^{-1}[\frac{1}{\sigma_w^2}V^\top V + \mathbb{E}(U_i^\top U_i)^{-1}]^{-1}\alpha}{\mathbb{E}(T_i^2) - \mathbb{E}(T_iU_i)[(\frac{1}{\sigma_w^2}V^\top V)^{-1} + \mathbb{E}(U_i^\top U_i)]^{-1}\mathbb{E}(U_i^\top T_i)}
\end{aligned}$$

The last equality once again follows from Sherman–Morrison–Woodbury formula. $\qquad\square$

*Proof for Corollary 1.1.* $\|(V^\top V)^{-1}\| = 1/\sigma^2_{\min}(V) \to 0$ so $\|[\frac{1}{\sigma^2_w}V^\top V + \mathbb{E}(U_i^\top U_i)^{-1}]^{-1}\| \to 0$. On the other hand, by Sherman–Morrison–Woodbury formula,

$$\mathbb{E}(T_iU_i)[(\frac{1}{\sigma^2_w}V^\top V)^{-1} + \mathbb{E}(U_i^\top U_i)]^{-1}\mathbb{E}(U_i^\top T_i)$$

$$= \mathbb{E}(T_iU_i)\mathbb{E}(U_i^\top U_i)^{-1}\mathbb{E}(U_i^\top T_i) - \mathbb{E}(T_iU_i)\mathbb{E}(U_i^\top U_i)^{-1}[\mathbb{E}(U_i^\top U_i) + \frac{1}{\sigma^2_w}V^\top V]^{-1}\mathbb{E}(U_i^\top U_i)^{-1}\mathbb{E}(U_i^\top T_i)$$

So the denominator term satisfies that

$$\mathbb{E}(T_i^2) - \mathbb{E}(T_iU_i)[(\frac{1}{\sigma^2_w}V^\top V)^{-1} + \mathbb{E}(U_i^\top U_i)]^{-1}\mathbb{E}(U_i^\top T_i) > \mathbb{E}(T_i^2) - \mathbb{E}(T_iU_i)[\mathbb{E}(U_i^\top U_i)]^{-1}\mathbb{E}(U_i^\top T_i)$$

which is bounded away from 0 by Lemma 7. Therefore, the asymptotic bias term (1) diminishes to 0. $\qquad\square$

**Proposition 2.** *Given the true $V$, $U_i$ can be estimated by the OLS estimator for the following linear regression: for $j = 1, \ldots, p$*

$$X_{ij} = V_jU_i^\top + \eta_{ij}.$$

*Namely we regress $X_i^\top$ on the design matrix $V$ to estimate $U_i^\top$. The resulting confounder estimator is $\tilde{U} = XV(V^\top V)^{-1}$. The subsequent OLS estimator for ATE based on $\tilde{U}$, $Y$ and $T$ is denoted as $\hat{\tau}$. Under the assumptions in Proposition 1, $\tilde{\tau}$ has the same asymptotic bias as in Proposition 1.*

*Proof of Proposition 2.* In this case, $\tilde{U} = U + WV(V^\top V)^{-1}$. Moreover, $\tilde{\tau}$ has the following form:

$$\hat{\tau} = [\frac{1}{N}T^\top(I - P_{\tilde{U}})T]^{-1}[\frac{1}{N}T^\top(I - P_{\tilde{U}})Y]$$

$$= [\frac{1}{N}T^\top(I - P_{\tilde{U}})T]^{-1}[\frac{1}{N}T^\top(I - P_{\tilde{U}})U\alpha + \frac{\tau}{N}T^\top(I - P_{\tilde{U}})T + \frac{1}{N}T^\top(I - P_{\tilde{U}})\epsilon]$$

Take $\frac{1}{N}T^\top P_{\tilde{U}}U\alpha$ as an example.

$$\frac{1}{N}T^\top P_{\tilde{U}}U\alpha = \frac{1}{N}T\tilde{U}(\frac{1}{N}\tilde{U}^\top\tilde{U})^{-1}\frac{1}{N}\tilde{U}^\top U\alpha$$

where

$$\frac{1}{N}T\tilde{U} = \frac{1}{N}T^\top[U + WV(V^\top V)^{-1}] \to \mathbb{E}T_iU_i$$

$$(\frac{1}{N}\tilde{U}^\top\tilde{U})^{-1} = \{\frac{1}{N}[U + WV(V^\top V)^{-1}]^\top[U + WV(V^\top V)^{-1}]\}^{-1}$$

$$\to \frac{1}{\sigma^2_w}[\frac{1}{\sigma^2_w}\mathbb{E}U_i^\top U_i + (V^\top V)^{-1}]^{-1}$$

$$\frac{1}{N}\tilde{U}^\top U\alpha = \frac{1}{N}[U + WV(V^\top V)^{-1}]^\top U\alpha \to \mathbb{E}U_i^\top U_i\alpha$$

Therefore

$$\frac{1}{N}T^\top P_{\tilde{U}}U\alpha \to \frac{1}{\sigma^2_w}\mathbb{E}T_iU_i[\frac{1}{\sigma^2_w}\mathbb{E}U_i^\top U_i + (V^\top V)^{-1}]^{-1}\mathbb{E}U_i^\top U_i\alpha$$

which is exactly equal to the limit of $\frac{1}{N}T^\top P_X U\alpha$ in the proof of Proposition 1. The equivalence of other terms can be verified similarly. $\qquad\square$

## A.2 Proof of Theorem 1

*Proof of Theorem 1.* The error of the ATE estimator in the linear regression can be written as:

$$\hat{\tau} - \tau = [\frac{1}{N}T^\top(I - P_{\hat{U}})T]^{-1}[\frac{1}{N}T^\top(I - P_{\hat{U}})U]\alpha + [\frac{1}{N}T^\top(I - P_{\hat{U}})T]^{-1}[\frac{1}{N}T^\top(I - P_{\hat{U}})\epsilon] \quad (7)$$

We first bound $\frac{1}{N}[T^\top(I - P_{\hat{U}})U]\alpha$:

$$\frac{1}{N}|T^\top(I - P_{\hat{U}})U\alpha| = \frac{1}{N}|T^\top(I - P_{\hat{U}})U\alpha - T^\top(I - P_U)U\alpha|$$

$$= \frac{1}{N}|T^\top(P_U - P_{\hat{U}})U\alpha|$$

$$\leq \frac{1}{\sqrt{N}}\|T\|\frac{1}{\sqrt{N}}\|U\alpha\|\|P_U - P_{\hat{U}}\|$$

$$\leq (\frac{2}{\sqrt{N}}\|T\|)(\frac{A}{\sqrt{Nr}}\|U\|)(r\angle(\hat{U}, U))$$

The first equaility follows from $(I - P_U)U = 0$. The last inequality follows from Lemma 4.

We then bound $[T^\top(I - P_{\hat{U}})T]$:

$$\frac{1}{N}|T^\top(I - P_{\hat{U}})T| = \frac{1}{N}|T^\top(I - P_U)T + T^\top(P_U - P_{\hat{U}})T|$$

$$\geq \frac{1}{N}T^\top(I - P_U)T - |\frac{1}{N}T^\top(P_U - P_{\hat{U}})T|$$

$$\geq \frac{1}{N}T^\top(I - P_U)T - \frac{2}{N}\|T\|^2\angle\Theta(U, \hat{U})$$

Furthermore, we can bound $\frac{1}{N}|T^\top(I - P_{\hat{U}})\epsilon|$: $T^\top(I - P_{\hat{U}})\epsilon$ is sub-Gaussian with mean 0 and variance $\sigma^2 T^\top(I - P_{\hat{U}})T$. By Hoeffding bound, for any $t > 0$ and some constant $c > 0$,

$$P(\frac{1}{n}|T^\top(I - P_{\hat{U}})\epsilon| \geq t) \leq 2e^{-\frac{cN^2t^2}{\sigma^2 T^\top(I - P_{\hat{U}})T}} \leq 2e^{-\frac{cNt^2}{\sigma^2}}$$

Take $t = \frac{\sigma}{N^{1/4}}$, then $\frac{1}{N}|T'(I - P_{\hat{U}})\epsilon| \leq \frac{\sigma}{N^{1/4}}$ with high probability $1 - 2\exp(-cN^{1/2})$ for some positive constant $c$.

Plug these three bounds in (7) then the conclusion follows. $\qquad\square$

**Lemma 4** (Equivalence of Space Distance Metrics). *The metric $\angle(\hat{M}, M)$ for matrices $M \in \mathbb{R}^{N \times r}$ and $\hat{M} \in \mathbb{R}^{N \times k}$ with orthonormal columns satisfies:*

$$\angle(\hat{M}, M) \leq \|\hat{M}\hat{M}^T - MM^T\| \leq 2\angle(\hat{M}, M)$$

*Proof.* See Lemma 1 in Cai et al. [31]. $\qquad\square$

**Lemma 5.** *Suppose that $T_i$ is almost surely not a linear combination of $U_i$. Under Assumption 5, $\frac{1}{N}T^\top(I - P_U)T$ is almost surely bounded away from 0 for any $N$.*

*Proof.* Consider the asymptotic case when $N \to \infty$.

$$\frac{1}{N}T^\top(I - P_U)T = \frac{1}{N}\sum_{i=1}^N T_i^2 - \frac{1}{N}\sum_{i=1}^N T_i U_i^\top(\frac{1}{N}\sum_{i=1}^N U_i U_i^\top)^{-1}\frac{1}{N}\sum_{i=1}^N U_i T_i$$

By Law of Large Number,

$$\frac{1}{N}\sum_{i=1}^N T_i^2 \to \mathbb{E}(T_i^2)$$

$$\frac{1}{N}\sum_{i=1}^N T_i U_i^\top(\frac{1}{N}\sum_{i=1}^N U_i U_i^\top)^{-1}\frac{1}{N}\sum_{i=1}^N T_i U_i \to \mathbb{E}(T_i U_i^\top)[\mathbb{E}U_i U_i^\top]^{-1}\mathbb{E}(U_i T_i)$$

The result follows immediately from a matrix version of Cauchy-Schwartz Inequality [37]. $\qquad\square$

## A.3  Proof of Theorem 2

**Lemma 6.** *Assume that $\Phi_{N \times p}$ is a low-rank matrix of rank atmost $r \ll N, p$. Further assume $\forall (i, j)$, $X_{ij} - g(\Phi_{ij})$ are sub-exponential with parameter $\sigma'$ and $|\Omega| > c_0 r \bar{N} \log \bar{N}$ for large enough constant $c_0$. Given any $\beta$ there exist positive constants $c_\beta, C_\beta, K_\beta$ such that for $\lambda = 2 c_\beta \sigma' \sqrt{N p} \sqrt{\frac{r \bar{N} \log \bar{N}}{|\Omega|}}$, the estimator from Exponential Family Matrix Completion (2) satisfies the following with probability at least $1 - 4 e^{-(1+\beta) \log^2 \bar{N}} - e^{-(1+\beta) \log \bar{N}}$:*

$$\|\hat{\Phi} - \Phi\|_F^2 \leq C_\beta \frac{\alpha_{sp}^2(\Phi) \max\{\sigma'^2, 1\}}{\mu_\beta^2} \left( \frac{r \bar{N} log \bar{N}}{|\Omega|} \right) \|\Phi\|_F^2 \tag{8}$$

*where $\mu_\beta = K_\beta e^{-\frac{2 \eta \alpha_{sp}(\Phi)}{\sqrt{N p}}} > 0$ for some positive constant $K_\beta$ and $\alpha_{sp}(\Phi)$ is the spikeness ratio of $\Phi$ defined as follows:*

$$\alpha_{sp}(\Phi) = \frac{\|\Phi\|_{\max} \sqrt{N p}}{\|\Phi\|_F}$$

*Proof.* See Corollory 1 in Gunasekar et al. [8] for sub-Gaussian $X_{ij} - g(\Phi_{ij})$. For sub-Exponential case, use the Orlicz norm corresponding to sub-Exponential random variables for Lemma 3 and Lemma 5 in Gunasekar et al. [8] and then the same conclusion follows.  $\square$

**Lemma 7** (Wedin's Theorem). *Suppose that $X = U \Sigma V^\top$ is of rank $r$ and $\hat{X} = X + E$ with the leading $r$ left singular vector matrix and right singular vector matrix being $\hat{U}$ and $\hat{V}$. Then*

$$\max\{\angle(\hat{U}, U), \angle(\hat{V}, V)\} \leq \frac{\max\{\|R\|_2, \|S\|_2\}}{\sigma_r(\hat{X})} \wedge 1$$

*where*

$$R = X \hat{V} - \hat{U} \hat{\Sigma} = -E \hat{V}$$
$$S = X' \hat{U} - \hat{V} \hat{\Sigma} = -E' \hat{U}$$

*Proof of Theorem 2.* Obviously $\frac{\alpha_{sp}(\Phi)}{\sqrt{N p}} < 1$, so $\mu_\beta > K_\beta e^{-2\eta}$. Let $c_2^2 = \frac{C_\beta \max\{\sigma'^2, 1\}}{K_\beta^2 e^{-4\eta}}$ with $\beta = 1$, then according to Lemma 6, the following holds with high probability at least $1 - 4 e^{-2 \log^2 \bar{N}} - e^{-2 \log \bar{N}}$:

$$\|\hat{\Phi} - \Phi\|_F^2 \leq c_2^2 \alpha_{sp}^2(\Phi) \frac{r \bar{N} log \bar{N}}{|\Omega|} \|\Phi\|_F^2$$

Apply Wedin's Theorem (Lemma 7) on $\Phi$ and $\hat{\Phi}$ with $E = \Phi - \hat{\Phi}$. Since $\hat{U}$ and $\hat{V}$ both have orthonormal columns,

$$\|R\|_2 \leq \|E\|_2 \leq \|E\|_F$$

$$\|S\|_2 \leq \|E\|_2 \leq \|E\|_F$$

where

$$\|E\|_F \leq c_2 \alpha_{sp}(\Phi) \sqrt{\frac{r \bar{N} \log \bar{N}}{|\Omega|}} \|\Phi\|_F$$

By Weyl's inequality,

$$\sigma_r(\hat{\Phi}) \geq \sigma_r(\Phi) - \|E\|_2 \geq \sigma_r(\Phi) - \|E\|_F$$

As a result,

$$
\begin{aligned}
\angle(\hat{U}, U) &\le \frac{c_2 \alpha_{sp}(\Phi)\sqrt{\frac{r\bar{N}\log\bar{N}}{|\Omega|}}\|\Phi\|_F}{\sigma_r(\Phi) - c_2\alpha_{sp}(\Phi)\sqrt{\frac{r\bar{N}\log\bar{N}}{|\Omega|}}\|\Phi\|_F} \wedge 1 \\[2mm]
&\le \frac{c_2 \alpha_{sp}(\Phi)\sqrt{\frac{r\bar{N}\log\bar{N}}{|\Omega|}}r\|\Phi\|_2}{\sigma_r(\Phi) - c_2\alpha_{sp}(\Phi)\sqrt{\frac{r\bar{N}\log\bar{N}}{|\Omega|}}r\|\Phi\|_2} \wedge 1 \\[2mm]
&\le \frac{c_2 \alpha_{sp}(\Phi)\sqrt{\frac{r^3\bar{N}\log\bar{N}}{|\Omega|}}}{\frac{\sigma_r(\Phi)}{\sigma_1(\Phi)} - c_2\alpha_{sp}(\Phi)\sqrt{\frac{r^3\bar{N}\log\bar{N}}{|\Omega|}}} \wedge 1
\end{aligned}
$$

$\square$

### A.4  Proof of Theorem 3

*Proof of Theorem 3.* The conclusions immediately follow from Lemma 8, Lemma 9, Lemma 10, Theorem 1, and Theorem 2. $\square$

**Lemma 8** (Spikeness Ratio). *Under Assumption 5, the spikeness ratio $\alpha_{sp}(\Phi) \le c' c_V \sqrt{r}$ with high probability $1 - N^{-1/2} - 2\exp(-cN^{1/2})$ for some positive constant $c, c'$.*

*Proof.* According to the definition, $\alpha_{sp}(UV^\top) = \sqrt{Np}\frac{\|UV^\top\|_{\max}}{\|UV^\top\|_F}$. Obviously,

$$\|UV^\top\|_{\max} \le \max_{ij}(U_i^T V_j) \le \max_i \|U_i\| \max_j \|V_j\|$$

Next, we prove that $\|UV^\top\|_F \ge \sigma_r(U)\|V\|_F$. Suppose $U$ has SVD $\bar{U}_{n\times r}\bar{\Sigma}_{r\times r}\bar{V}_{r\times r}^T$ where $\bar{\Sigma}_{r\times r} = \mathrm{diag}(\sigma_1(U),...,\sigma_r(U))$ and $\bar{V}^T\bar{V} = \bar{V}\bar{V}^T = I_{r\times r}$. Then

$$
\begin{aligned}
\|UV\|_F &= \|\bar{U}_{n\times r}\bar{\Sigma}_{r\times r}\bar{V}_{r\times r}^T V\|_F \\[2mm]
&= \left\| \begin{bmatrix} \sigma_1(U) & 0 & \dots & 0 \\ 0 & \sigma_2(U) & \dots & 0 \\ \vdots & \vdots & \ddots & \vdots \\ 0 & 0 & \dots & \sigma_r(U) \end{bmatrix} \begin{bmatrix} \bar{V}_1^T V \\ \bar{V}_2^T V \\ \vdots \\ \bar{V}_r^T V \end{bmatrix} \right\|_F \\[2mm]
&= \sqrt{\sum_{k=1}^r \|\sigma_k(U)\bar{V}_k^T V\|^2} \\[2mm]
&\ge \sigma_r(U)\sqrt{\sum_{k=1}^r \|\bar{V}_k^T V\|^2} \\[2mm]
&= \sigma_r(U)\|\bar{V}^T V\|_F = \sigma_r(U)\|V\|_F
\end{aligned}
$$

Therefore, $\alpha_{sp}(UV^\top) \le \sqrt{Np}\frac{\frac{1}{\sqrt{N}}\max_i\|U_i\|\max_j\|V_j\|}{\frac{1}{\sqrt{N}}\sigma_r(U)\|V\|_F}$. Following the similar proof in Lemma 10, we can prove that the following holds with high probability at least $1 - 2\exp(-cN^{1/2})$:

$$\frac{1}{\sqrt{N}}\sigma_r(U) \ge (1 - C\sqrt{\frac{r}{N}} - \frac{1}{N^{1/4}})\|L\|$$

Under Assumption 5, $\|L^{-1}U_i\|^2 \sim \chi^2(r)$. Then according to Proposition 1 in [39], with probability at least $1 - \exp(-t^2/2)$, $\|U_i\| \le \sqrt{r + t\sqrt{2r} + t^2} \le \sqrt{r} + t$. Let $t = \sqrt{3\log N}$ and take union bound over $i = 1, ..., N$, then with high probability $1 - N^{-1/2}$ for any $i$,

$$\frac{1}{\sqrt{N}}\|U_i\| \le (\frac{\sqrt{r}}{\sqrt{N}} + \frac{\sqrt{3\log N}}{\sqrt{N}})\|L\|$$

which implies that $\frac{1}{\sqrt{N}}\max_i \|U_i\| \leq (\frac{\sqrt{r}}{\sqrt{N}} + \frac{\sqrt{3\log N}}{\sqrt{N}})\|L\|$.

Therefore, with high probability $1 - N^{-1/2} - 2\exp(-cN^{1/2})$,

$$\alpha_{sp}(UV^\top) \leq \frac{\sqrt{r} + \sqrt{3\log N}}{1 - C\frac{\sqrt{r}}{\sqrt{N}} - \frac{1}{N^{1/4}}}\sqrt{p}\frac{\max_j \|V_j\|}{\|V\|_F} \leq c' c_V \sqrt{r}$$

$\square$

**Lemma 9.** *Under Assumption 5, $\frac{\sigma_r(\Phi)}{\sigma_1(\Phi)} \geq \sqrt{\frac{\underline{v}}{\underline{v}+2\overline{v}}}$ with high probability $1 - 2\exp(-Cp^\delta)$ given that $p^{1+\delta}/N \to 0$ for some positive constant $\delta, C$.*

*Proof.* We aim to prove $|x^\top(\frac{1}{N}VU^\top UV^\top - VL^\top LV^\top)x| \leq \epsilon$ for any $x$ on the $p$-dimensional unit sphere $\mathcal{S}^{p-1}$. Since $x^\top(\frac{1}{N}VU^\top UV^\top - VL^\top LV^\top)x = 0$ for $x \in \mathrm{Null}(V)$, we only have to prove

$$\max_{x\in\mathcal{S}^{p-1}\cap\mathrm{Null}^\perp(V)} |x^\top(\frac{1}{N}VU^\top UV^\top - VL^\top LV^\top)x| \leq \epsilon$$

where $S^{p-1} \cap \mathrm{Null}^\perp(V)$ is a $r$-dimensional space.

Consider $\frac{1}{4}$-net $\mathcal{N}$ for $\mathcal{S}^{D-1} \cap \mathrm{Null}^\perp(V)$, according to Lemma 5.4 in [38],

$$\max_{x\in\mathcal{S}^{p-1}\cap\mathrm{Null}^\perp(V)} |x^\top(\frac{1}{N}VU^\top UV^\top - VL^\top LV^\top)x| \leq 2\max_{x\in\mathcal{N}} |x^\top\frac{1}{N}VU^\top UV^\top x - x^\top VL^\top LV^\top x|$$

So we only need to prove that $\max_{x\in\mathcal{N}} |x^\top\frac{1}{N}VU^\top UV^\top x - x^\top VL^\top LV^\top x| \leq \frac{\epsilon}{2}$ with high probability. Note that $\frac{1}{N}x^\top VU^\top UV^\top x - x^\top VL^\top LV^\top x = \frac{1}{N}\sum_i (Z_i^2 - \mathbb{E}(Z_i^2))$, where $Z_i = U_i V^\top x$ are mutually independent with $\mathbb{E}(Z_i) = 0$ and $\mathbb{E}(Z_i^2) = x^T VL^\top LV^\top x \leq \|VL^\top\|^2$. It follows that the $Z_i^2 - \mathbb{E}Z_i^2$ are sub-Exponential with upper bounded sub-Exponential norm (Lemma 5.14 [38]):

$$\|Z_i^2 - \mathbb{E}Z_i^2\| \leq \|Z_i^2\|_{\psi_1} + \mathbb{E}Z_i^2 \leq 2\|Z_i\|_{\psi_2}^2 + \mathbb{E}Z_i^2 \leq 3\|VL\|^2$$

By the Berstein Inequality (Corollary 5.17 in [38])

$$\mathbb{P}(|x'(\frac{1}{N}VU^\top UV^\top - VL^\top LV^\top)x| \geq \frac{\epsilon}{2}) \leq 2\exp(-c\min\{\frac{\epsilon}{6\|VL\|}, \frac{\epsilon^2}{36\|VL\|^2}\}N)$$

Furthermore, Lemma 5.2 in [38] implies that $|\mathcal{N}| \leq 9^r$. So taking union bound over $\mathcal{N}$ gives:

$$\mathbb{P}(\max_{x\in\mathcal{N}} |x'(\frac{1}{N}VU^\top UV^\top - VL^\top LV^\top)x| \geq \frac{\epsilon}{2}) \leq 2\exp(r\log 9 - \frac{c}{\tilde{K}}\min\{\epsilon, \epsilon^2\}N)$$

where $\tilde{K}^{-1} = \min\{\frac{1}{6\|VL\|^2}, \frac{1}{36\|VL\|^4}\}$.

We consider two cases:

1. For large enough $p$ ($6\underline{v}p > 1$), take $\epsilon = \frac{p^{2+\delta}}{N}$, then for some positive constant $C$ and $r/p^\delta \to 0$,

   $$\mathbb{P}(\max_{x\in\mathcal{N}} |x^\top(\frac{1}{N}VU^\top UV^\top - VL^\top LV^\top)x| \geq \frac{\epsilon}{2}) \leq 2\exp(r\log 9 - \frac{c}{36\underline{v}^2 p^2}\epsilon N) \leq 2\exp(-Cp^\delta)$$

   So with probability at least $1 - 2\exp(-Cp^\delta)$,

   $$\frac{\sigma_r^2(UV^\top)}{\sigma_1^2(UV^\top)} \geq \frac{\sigma_r^2(VL^\top) - \epsilon}{\sigma_1^2(VL^\top) + \epsilon} \geq \frac{\underline{v} - \epsilon/p}{\overline{v} + \epsilon/p} = \frac{\underline{v} - \frac{p^{1+\delta}}{N}}{\overline{v} + \frac{p^{1+\delta}}{N}} \geq \frac{\underline{v}}{2\overline{v} + \underline{v}}$$

   which is bounded away from 0 for large enough $N, p$ such that $\frac{p^{1+\delta}}{N} \leq \frac{\underline{v}}{2}$.

2. For moderate $p$ ($6\underline{v}p \leq 1$), take $\epsilon = \frac{p^{1/2+\delta/2}}{N}$ and then

   $$\mathbb{P}(\max_{x\in\mathcal{N}} |x'(\frac{1}{N}V^T U^T UV - V^T V)x| \geq \frac{\epsilon}{2}) \leq 2\exp(r\log 9 - \frac{c}{6\underline{v}p}\epsilon^2 N) \leq 2\exp(-Cp^\delta),$$

   which implies the same conclusion.

$\square$

**Lemma 10.** *Under Assumption 5, $\frac{1}{\sqrt{Nr}}\|U\|$ is bounded for any $N$ with high probability at least $1 - 2\exp(-cN^{1/2})$.*

*Proof.* Apply Theorem 5.39 in [38] to matrix $L^{-1}U$, for any $t > 0$ and positive constants $c, C$, with probability at least $1 - 2\exp(-ct^2)$,

$$\frac{1}{\sqrt{Nr}}\|U\| \le \frac{1}{\sqrt{Nr}}\|UL^{-1}L\| \le (1 + C\frac{\sqrt{r}}{\sqrt{N}} + \frac{t}{\sqrt{N}})\frac{1}{\sqrt{r}}\|L\|$$

Take $t = N^{1/4}$ then the conclusion follows.

$\square$

## B  More Numerical Results

Figure 4: Relative RMSE for binary covariates in the low dimensional setting as in Section 4.1 and the relative RMSE for the setting where $p$ varies from 150 to 1500 and $N = p/1.5$.

Figure 5: Relative RMSE of ATE estimators for binary covariates with $N = 200, 400, \dots, 2000$ and $p = N/2$. Each entry is set to be missing value with equal probability 0, 0.3, or 0.5.

## C  Causal Effect Variational Autoencoder

Figure 7 shows the estimation error of the causal effect variational autoencoder (CEVAE) [9] in the twins dataset when there are no missing values. Like Exponential Family Matrix Completion, variational autoencoder (VAE) is another latent variable model that provides yet another way to recover the latent confounders. In [9], the authors combine such a latent variable model with an outcome and treatment model and train these together in order to recover causal effects in the presence of noisy proxies. This flexible neural-net-based model allows for additional non-linearities that provide for better performance in this semi-synthetic example, as we explain below. Indeed, these benefits disappear when we limit the outcome model to be linear.

Figure 6: Relative RMSE of ATE estimators for Gaussian and Binary covariates with $N = 150, 300, \ldots, 1500$ and $p = 200$.

Figure 7: Performance of causal effect variational autoencoder [9] on the twins dataset without missingness. CEVAE follows the neural network architecture given in [9], and CEVAE Linear uses 0 hidden layer in $\mathbb{P}(Y_i \mid T_i, U_i)$ while keeping the archetecture of other neural networks the same as CEVAE.

In our twins data example, the proxies are synthetic: we replicate the GESTAT10 multiple times and independently perturb the entries of these copies with probability $0.5$; each perturbed entry is then assigned with a new value sampled from $0$ to $9$ uniformly at random. This means that for $a \in \{0, \ldots, 9\}$, $P(X_{ij} = a \mid U_i) = 0.5 \times \frac{1}{10} + 0.5 \times \mathbb{I}(U_i = a)$. In contrast, the matrix factorization method based on multinomial loss assumes that $P(X_{ij} = a \mid U_i) = \exp(U_i^\top V_j^a) / \sum_a \exp(U_i^\top V_j^a)$, where $V_j^0, \ldots, V_j^9$ define the complete set of loading vectors of the $j^{th}$ noisy proxy [18]. Although this assumption drastically deviates from the true proxy generating process, using matrix factorization still leads to considerable improvement in causal effect estimation. Thanks to its high non-linearity, CEVAE can even better adapt to this complex synthetic emission model and learn it more faithfully, therefore producing the better results see in Figure 7. While CEVAE has no theoretical guarantees, our work focuses on providing the first finite-sample recovery result for a causal parameter from high-dimensional proxy data, which is only possible in the simpler linear setting. Indeed, our work can be seen as providing some theoretical justification for more practical methods using more complex models.

To further study where the benefit of CEVAE stems from, we replaced the outcome model with a linear one (i.e., no hidden layers, essentially a logistic regression) and found that the performance deteriorated significantly (CEVAE Linear in Figure 7). This indeed suggests that the primary improvement arises from the high flexibility of the neural networks in CEVAE. In fact, while logistic regression on the matrix-factorization-recovered confounders improves significantly on simple logistic regression, CEVAE with a linear outcome does as badly as simple logistic regression when the number of proxies is large.