[Reviews · NeurIPS 2018]

Reviewer 1



This paper focuses on the problem of causal inference in the presence of confounders. Particularly, the authors propose to use matrix factorization to infer the confounders from noisy covariates. Then the recovered confounders are used in the causal effect estimation. With this correction, the authors show that the asymptotic bias diminishes to 0 when ||V|| approaches to infinity. Quality: This paper tries to solve an important problem in causal inference: to reduce estimation bias in the presence of noisy covariates. The overall presentation is ok and the proposed method is reasonable. Clarity: Overall, the presentation is ok. In Assumption 2, the authors take the low rank assumption for granted. It is better to give some intuitions for the specific scenario in the paper. Originality: The proposed method relies on low rank matrix factorization, which has been well studied. But from my knowledge, its application in causal inference is novel. A related paper is cited in [9], which makes use of variational autoencoders. Significance: Reducing the estimation bias of causal effect is an interesting problem to explore. The proposed method is reasonable. However, the authors did not compare it with the state-of-the-art method, which is proposed in [9]. If the comparison was made, the efficacy of the proposed method could be better demonstrated.

Reviewer 2



In this paper, the authors use matrix factorization to reconstruct latent confounders from their noisy proxies, tolerating also the case in which the proxy measurements are missing at random. Focusing on linear regression, and given some extra assumptions, they provide an error bound for ATE after this preprocessing step, and show the consistency of the estimator after preprocessing. To the best of my knowledge, this paper presents an interesting idea with a thorough theoretical analysis. In my opinion, the paper is quite dense and technical, making it quite difficult to read for a causality reader. The clarity could be improved, especially regarding the several assumptions used in different parts of the paper, some of which are vary in the different theorems. The results on both synthetic and real data seem promising. A possible flaw is that this paper does not mention much related work in either reconstructing latent variables or in general estimating causal effects with unmeasured confounders, e.g. https://arxiv.org/abs/1609.08816. Minor comments; Typos: marix (l74), constrast (l75), corrolary 1.1 (l148), backpropogation (l197), guanrantee (l245), numebr (l237) L114: e_i and e_j are not defined L175 (assumption 4): h is not defined L200: unclear phrasing

Reviewer 3



Summary: The paper describes a novel method to compute average causal treatment effects via adjustment when a large number of noisy proxy covariates is available. It treats the (many) proxy covariates as noisy observations (measurement noise incl. missing values) of a few unobserved/underlying ‘true’ confounders that should ideally be used for adjustment. It then proposes to use low-rank matrix factorization as a preprocessing step on the many observed noisy covariates to uncover / reconstruct a much better approximation to these true confounders. It is subsequently shown on both synthetic and real-world data sets that using these low rank constructs as input to existing methods to compute treatment effects for complete/missing data significantly improves their performance. Frequently encountered problem, very interesting idea, very well executed. Although I am aware that many approaches using latent class/factor analysis etc. on this type of data exist, to the best of my knowledge the proposed method and combination with ATE estimates is novel, with promising results and should be relevant to a wide audience. As far as I can see there are no significant flaws in the theoretical results. Especially the bounds on the treatment effect estimates are impressive, although I did not check the derivation of eq.(3) and (4) in detail. One point of concern is that the method effectively assumes that the low-rank confounder constructs are independent: if we assume these correspond to psychometric factors like intelligence and personality traits (as in the examples provided), then this is probably unlikely to be true. However, in many other cases it may be sufficiently reasonable, and I suspect that even when it does not hold then it will still help to obtain a better ATE than using the myriad of noisy covariates as a starting point for adjustment. The paper as a whole is well written, and good to follow, although it tends to get quite technical at some points, and might benefit from 1 or 2 explanatory figures. In summary: clear accept. ======= Details: 74: typo ‘marix’ 69-81: bit compact for the large body of related work 97: ‘.. treatment and control respectively’ => this suggest 0=treatment, 1=control … 115+ : perhaps a small figure depicting system + model would be nice here … 149: ‘.. so long as the .. not vanish too fast’ => given a concrete example of where this is the case 169: MCAR is usually a rather strong/unrealistic assumption in psychometric tests, but ok … 172: ‘.. exp.family noise mechanism’. => does this relate to the measurement noise or the intrinsic noise? 175: typo ‘anlytic’ 180: give some intuition on the purpose/implication of this assumption on \grad^2G(u) 192: typo ‘matirx’ 204: typo: ’an error bound’ or ‘error bounds’ 237: typo ‘numebr’ idem: try to give a concrete example where this is the case 246: nice bound … though I could not check this derivation :) Fig.1 : caption should be a bit more informative 260-269,eq.(5): how would you solve this for a combination of continuous and discrete covariates simultaneously? 338/Fig2: good & interesting results … can you explain the dramatic difference between Match(ing) with and without the MF preprocessing step? (initially without seems to get worse for increasing nr. of proxies?) idem: make caption more useful